# Predictors of Sarcopenia in Outpatients with Post-Critical SARS-CoV2 Disease. Nutritional Ultrasound of Rectus Femoris Muscle, a Potential Tool

**DOI:** 10.3390/nu14234988

**Published:** 2022-11-24

**Authors:** Isabel Cornejo-Pareja, Ana Gloria Soler-Beunza, Isabel María Vegas-Aguilar, Rocío Fernández-Jiménez, Francisco J. Tinahones, Jose Manuel García-Almeida

**Affiliations:** 1Instituto de Investigación Biomédica de Málaga (IBIMA), Virgen de la Victoria University Hospital, 29010 Málaga, Spain; 2Centro de Investigacion Biomedica en Red de la Fisiopatología de la Obesidad y Nutricion (CIBEROBN), Instituto de Salud Carlos III (ISCIII), 29010 Malaga, Spain; 3Department of Endocrinology and Nutrition, Arnau de Villanova University Hospital, 25198 Lleida, Spain; 4Department of Endocrinology and Nutrition, Virgen de la Victoria University Hospital, Málaga University, 29010 Málaga, Spain

**Keywords:** sarcopenia, predictors of sarcopenia, ultrasound of rectus femoris muscle, post-critical SARS-CoV2 disease, COVID-19 survivors, rectus femoris

## Abstract

Background and Objectives: The loss of muscle mass in post-critical COVID-19 outpatients is difficult to assess due to the limitations of techniques and the high prevalence of obesity. Ultrasound is an emerging technique for evaluating body composition. The aim is to evaluate sarcopenia and its risk factors, determining ultrasound usefulness as a potential tool for this purpose according to established techniques, such as the bioimpedance vector analysis (BIVA), handgrip strength, and timed up-and-go test. Methods: This is a transversal study of 30 post-critical COVID-19 outpatients. We evaluated nutritional status by ultrasound (Rectus Femoris-cross-sectional-area (RF-CSA), thickness, and subcutaneous-adipose-tissue), BIVA, handgrip strength, timed up-and-go test, and clinical variables during admission. Results: According to The European Society for Clinical Nutrition and Metabolism and the European Association for the Study of Obesity (ESPEN&EASO) Consensus for Sarcopenic and Obesity, in terms of excess fat mass and decreased lean mass, the prevalence of class-1 sarcopenic obesity was 23.4% (*n* = 7), and class-2 sarcopenic obesity was 33.3% (*n* = 10) in our study. A total of 46.7% (*n* = 14) of patients had a handgrip strength below the 10th percentile, and 30% (*n* = 9) achieved a time greater than 10s in the timed up-and-go test. There were strong correlations between the different techniques that evaluated the morphological (BIVA, Ultrasound) and functional measurements of muscle. Intensive care unit stay, mechanical ventilation, and age all conditioned the presence of sarcopenia in COVID-19 outpatients (R^2^ = 0.488, *p* = 0.002). Predictive models for sarcopenic diagnosis based on a skeletal muscle index estimation were established by RF-CSA (R^2^ 0.792, standard error of estimate (SEE) 1.10, *p* < 0.001), muscle-thickness (R^2^ 0.774, SEE 1.14, *p* < 0.001), and handgrip strength (R^2^ 0.856, SEE 0.92, *p* < 0.001). RF-CSA/weight of 5.3 cm^2^/kg × 100 was the cut-off value for predicting sarcopenia in post-critical COVID-19 outpatients, with 88.2 sensitivity and 69.2% specificity. Conclusion: More than half of the post-critical COVID-19 survivors had sarcopenic obesity and functional impairment of handgrip strength. Intensive care unit stay, age, and mechanical ventilation all predict sarcopenia. An ultrasound, when applied to the assessment of body composition in post-critical COVID-19 patients, provided the possibility of assessing sarcopenia in this population.

## 1. Introduction

Patients with acute COVID-19 infection tended to lose weight and cachexia in the context of a series of catabolic and metabolic alterations that directly affected their nutritional status [1]. Studies suggest that around 12% of these hospitalized patients ended up requiring a stay in intensive care units (ICU), which per se included a significant risk of malnutrition, affecting functionality and quality of life [2].

In recent years, techniques such as bioelectrical impedance vectorial analysis (BIVA) or ultrasound have played a significant role in evaluating the body composition of patients with malnutrition. BIVA is a technique that allows the characterization of changes in body composition (adipose tissue, muscle mass) based on the ability of the human body to transmit an electrical current. In addition, it provides bioelectric parameters, such as the phase angle (PhA), when calculated from the raw bioelectrical measurements (the reactance and resistance of the tissues), which is a global marker of both the nutritional and inflammatory status of the patient. In previous studies, it has been useful as an independent prognostic factor in a variety of pathologies, including in patients with acute COVID-19 infection [3].

Nutritional ultrasound^®^ (NU^®^) is a new concept that uses the determination of measurements by means of ultrasound to discriminate and evaluate the thickness of lean and fatty layers to extrapolate the body composition (lean mass and fat mass) of the organism. It includes the evaluation of body compartments: adipose, muscle, connective, vascular, and bone tissue, with ultrasound. It is an advanced clinical examination tool that is emerging as a method for evaluating the different body compartments, and there are different validation studies on the measurement technique [4]. Although different muscular structures can be evaluated, most of the studies focus on the rectus femoris (RF), one of the most referenced measures for its correlation with strength and tests of execution or functional performance [5,6,7].

Apart from being able to measure muscle morphology parameters, such as length, volume, and area, the assessment of echogenicity provides us with information on muscle architecture and quality, assessing items such as fat infiltration or muscle necrosis with precise image analysis techniques [8].

However, the specific role that COVID-19 plays in the loss of muscle strength and mass in these patients has not been comprehensively studied in this population, and there is little evidence about the sarcopenia prevalence, follow-up, monitoring techniques, and adequate nutritional support that these patients should receive after hospital discharge. The objective of our study was to evaluate sarcopenia using new morphofunctional evaluation techniques, as well as the predictive factors that may influence the development of sarcopenia in post-critical COVID-19.

## 2. Materials and Methods

### 2.1. Study Design

This is a transversal study carried out at the Virgen de la Victoria University Hospital (Malaga, Spain) from October 2021 to February 2022. The patients were evaluated consecutively in a specific nutrition medical office 15–30 days after hospital discharge; their admission was due to an acute COVID-19 infection that required an ICU stay. During admission, all patients were diagnosed with COVID-19 pneumonia according to the World Health Organization interim guidance (WHO) with SARS symptoms and were tested by nasopharyngeal samples at admission using real-time reverse transcriptase-polymerase chain reaction assays. All subjects gave their informed consent for inclusion before they participated in the study. The study was conducted in accordance with the Declaration of Helsinki, and the protocol was approved by the Ethics Committee of the Virgen de la Victoria (PI-20–321, September 2021). All patients included in our study met the inclusion criteria (previous ICU admission for COVID-19 pneumonia, agreed to participate in the study by accepted informed consent), and no exclusion criteria (participation declined or inability to measure by muscle ultrasound or BIVA: amputations, extensive skin lesions, local hematomas…). During their hospital stay and upon discharge, all patients received a protocolized program of nutritional support and functional rehabilitation with exercise recommendations for muscle recovery by specific strength exercises and a high-protein oral supplementation with about 40 g of extra protein. Exercises aimed at muscle strengthening are recommendations established in our center together with the Rehabilitation Unit. The recommendation was strength exercises with weights and elastic bands of progressive intensity until reaching fatigue three times per week and with a session duration of 30–45 min. We proposed both upper limb strength exercises (shoulder extension and row with elastic bands or shoulder hold and push-up arms with weights) and trunk and lower limb (trunk extension, abdominals, squats or leg and knee extensions, knee bend, tiptoe). A flow chart diagram shows the patient selection process for our study (Appendix A).

### 2.2. Body Composition Analysis

#### 2.2.1. Phase Angle by BIVA

We measured the bioelectrical impedance and applied the BIVA graph to the patients 15–30 days after hospital discharge. Whole-body BI measurements were obtained with a 50 kHz, phase-sensitive impedance analyzer (BIA 101 Whole Body Bioimpedance Vector Analyzer (AKERN, Pontassieve, Italy)) which introduces 800 mA using tetrapolar electrodes positioned on the right hand and foot. All patients waited five minutes in a supine position before obtaining their BI measurements. The body consists of complex circuits composed of resistive (R) and capacitive (Xc) elements that, when stimulated with an alternating current, experience a frequency-dependent delay in the current concerning the flow of voltage.

An individual, standardized phase angle (SPhA) value was determined from the gender and age-matched reference population value by subtracting the reference PhA value from the observed patient PhA and then dividing the result by the respective age and gender reference standard deviation (SD).

The bioelectric parameters to estimate the body composition were analyzed, such as fat mass (FM), fat mass index (FMI), fat mass/height (FM/h), fat-free mass (FFM), fat-free mass index (FFMI), body cell mass (BCM), body cell mass/height (BCM/h), skeletal muscle mass/weight (SMM/w), muscle mass (MM), appendicular skeletal muscle mass (ASMM), skeletal muscle index (SMI), and hydration percentage.

According to The European Society for Clinical Nutrition and Metabolism and the European Association for the Study of Obesity (ESPEN and EASO) Consensus Statement for Sarcopenic Obesity [9,10], obesity was defined by an FM percentage higher than 30 in males and 40% in females, respectively. FMI higher than a 1-standard deviation (SD) of the population reference levels (7.2 and 10.3 kg/m for male and female, respectively), and an FMI higher than a 2-SD of the population reference levels (8.4 and 12 kg/m for male and female, respectively). Lean body mass was also evaluated following the recommendations of the ESPEN and EASO Consensus Statement for Sarcopenic Obesity. Mild or class-1 sarcopenia could diagnose with a skeletal muscle mass (SMM)/weight below than 1-SD of the population reference levels (37 and 27.6 for males and females, respectively), and severe or class-2 sarcopenia with an SMM/weight lower than 2-SD of the population reference levels (31.5 and 22.1 for male and female, respectively) [9,10].

#### 2.2.2. Nutritional Ultrasound^®^

##### Rectus Femoris (RF) Ultrasound Assessment

We performed a thorough nutritional ultrasound assessment with a HITACHI ALOKA F37 ultrasound scanner with an Aloka UST-5413 Linear Array transducer with a frequency range of 5.0–10 MHz in B-mode in a transverse position (Hitachi Europe, Stoke Poges Ltd., Buckinghamshire, UK). During the assessment, the participants lay in a supine position with their arms supinated and knees extended and relaxed to full extension. This must be performed by qualified professionals in the area. The probe was coated with an adequate water-soluble transmission gel to provide acoustic contact without depression of the dermal surface and was aligned perpendicularly to the longitudinal and transversal axes of RF to obtain the transverse image. Three images were registered of the right RF muscle, and for greater accuracy, the averaged measurements were estimated. The acquisition site was located two-thirds of the way along the femur length, measured between the upper pole of the patella and the anterior superior iliac spine. We measured in the transversal axis the cross-sectional area (CSA) in cm^2^, muscle circumference in cm, muscle thickness (or Y-axis), and the X-axis in cm [8,11] Figure 1.

Muscle echogenicity was determined by the grey-scale histogram analysis of the images. The echo intensity was quantified with ImageJ software (National Institutes of Health, NIH, MD, Bethesda, MD, USA). We valued the average, minimum and maximum echo intensity (between 0 “black” and 255 pixels “white”). These results were also expressed considering the average of the three measurements. The largest free-form area devoid of artifacts was defined, then the median grey-scale value for the selected region was recorded. The images were reviewed by a second investigator who was not directly involved in the image acquisition Figure 2.

##### Adipose Tissue Ultrasound Assessment

The subcutaneous and visceral preperitoneal adipose tissue at the abdominal level was also assessed. The acquisition site was located midway along the anterior midline, which was measured between the xiphoid and the umbilicus, with the patient in the supine position. This must be performed by qualified professionals in the area. The images were taken during unforced expiration, in a transverse axis, and aligned perpendicularly to the skin. The superficial and deep adipose tissue layers were differentiated. The visceral adipose tissue was determined by measuring the distance between the limit of the parietal peritoneum to the inner face at the junction of the two rectus abdominal muscles. To minimize the variability of the measurement three measurements were made, and the mean value was noted [11,12] Figure 3.

### 2.3. Functional and Muscle Strength Assessment

Handgrip strength (HGS) was determined using the JAMAR-Dynamometer (J A Preston Corporation, New York, NY, USA). The dominant hand was tested. Three measurements were taken, and the average was reported and compared with the published population reference data that were used as cut-off points [13].

The timed up-and-go (TUG) test was used as an assessing function. To measure the TUG time of participants, a colored tape was marked 3 m away from an armless chair in which participants were sitting. Participants were asked to walk 3 m, turn around the marked tape, and return to the chair as fast as they could. A timer was set as soon as the patient stood up from the chair and was stopped when the patient was seated again. At least one practice trial was performed before the test [14].

### 2.4. Clinical Variables

We determined the following clinical assessments: age, sex, stay in ICU, corticotherapy dose, stress hyperglycemia, mechanical ventilation, prone maneuvers, and cardiovascular comorbidities (e.g., history of diabetes, hypertension, dyslipidaemia, obesity). 

### 2.5. Statistical Analysis

Statistical analyses of the data were primarily performed using the SPSS program (version 22.0.0 for Windows, SPSS Iberica, Spain). We used descriptive statistics to characterize our cohort of patients. Baseline characteristics were expressed as the mean (interquartile range) for continuous variables and as proportions for the categorical variables. Continuous variables were compared with Student’s T-test or the Mann–Whitney U test according to their distribution. Categorical variables were compared with the Chi-squared (or Fisher’s exact test). We also analyzed the relationship using Pearson or Spearman correlation models according to normal distribution. After the analyses, the parameters which contributed to the best explanation of the model were retained in the multiple linear regression.

A multiple linear regression analysis was used to establish a predictive model based on the clinical variables (ICU stay, mechanical ventilation, and age) for the estimation of sarcopenia, such as the decline in SMI in post-critical COVID-19 outpatients. In addition, predictive models were established based on clinical variables (age, stress, hyperglycaemia, dexamethasone treatment, fat mass, and TUG) to find the qualitative ultrasonographic characteristics of the muscle. Finally, a multiple linear regression analysis was used to establish a predictive model based on muscle mass by ultrasound (RF-CSA and muscle thickness) and HGS for the estimation of SMI; while entering age (y), BMI (Kg/m^2^), and gender into the regression and using a stepwise method. When gender was entered into the regression, values of 0 and 1 were assigned for males and females, respectively.

The evaluation of the diagnostic performance of rectus the femoris muscle mass by ultrasound was standardized by weight based on the receiver operating characteristic (ROC) curve and the area under the curve (AUC). We estimated the accuracy of sarcopenia in our population using AUC by constructing a plot of sensitivity versus 1-specifity. ROC curves were used to determine the optimum cut-off values.

## 3. Results

### 3.1. Baseline Characteristics

A total of 30 patients were admitted to the ICU for severe acute COVID-19 and were subsequently attended to a specific nutrition medical office. The baseline demographic and clinical characteristics of the study participants are included in Table 1. The median age of the post-critical COVID-19 patients was 60 ± 9.4 y, with a BMI of 31.6 ± 7.4 kg/m^2^. Predominantly, 76.7% were male, and 86.6% were overweight or obese, with 23.3% classified as overweight (BMI 25–29.99 kg/m^2^) and 63.3% as obese (BMI ≥ 30 kg/m^2^), which was related to a longer ICU stay (r = 0.438, *p* = 0.018), The presence of other metabolic comorbidities included dyslipidaemia (43.3%), arterial hypertension (53.3%) and, although only 26.7% of patients were previously diabetic, 73.3% presented stress hyperglycaemia during admission. A total of 53.3% of patients required mechanical ventilation, and 46.7% were in a prone position during their ICU stay, while a total of 53% of the post-critical COVID-19 patients required home oxygen therapy after hospital discharge. In addition, more than 80% of the patients required high doses of corticosteroid therapy during their admission. The median length of hospital stay was 23 ± 19.9 d.

Patients who required more aggressive therapies, such as mechanical ventilation or prone maneuvers, were associated with stress hyperglycaemia during admission [r = 0.494, *p* = 0.006] and [r = 0.413, *p* = 0.023], respectively, longer hospital stays [r = 0.707, *p* < 0.001] and [r = 0.506, *p* = 0.004], respectively, and more days in the ICU [r = 0.740, *p* < 0.001] and [r = 0.523, *p* = 0.003], respectively; and they presented a slower functional recovery related to a lower mean dynamometry [r = −414, *p* = 0.023] and [r = −0.383, *p* = 0.036], respectively.

### 3.2. Risk Factors of ICU Admission That Condition Muscle Mass (Sarcopenia)

In addition, multiple linear regression models were constructed to show the effects of clinical variables during admission on the muscle mass (sarcopenia) of post-critical COVID-19 outpatients. The length of ICU stays, the need for mechanical ventilation, and age were significant risk factors of sarcopenia (based on SMI) in post-critical COVID-19 outpatients (corrected R^2^ = 0.488, *p* = 0.002) Table 2.

### 3.3. BIVA Analysis and Body Composition Estimates for the Diagnosis of Sarcopenia and Excess Fat Mass

The median PhA of post-critical COVID-19 patients was 4.8° (4.3–5.6), while SPhA was −0.45 (−1.7–0.25). PhA, a global measure of the nutritional and inflammatory status of patients, was negatively related to the need to use mechanical ventilation (r = −0.371, *p* = 0.043) and positively correlated with a greater HGS assessed by dynamometry (r = 0.462, *p* = 0.01).

Body composition analysis by BIVA measurements revealed characteristics that related to the sarcopenic obesity phenotype. We observed that FM levels were increased with a total FM of 28.8 Kg (24.9–35.6) and were standardized by a height of 16 kg/m (12.5–20.1) for males and 20.9 Kg/m (16.5–37.5) for females. According to the ESPEN and EASO Consensus Statement for Sarcopenic Obesity [9,10], 76.7% of our population had one standard deviation (SD) higher FMI than the population reference levels, and 60% of patients had 2SD of FMI above the population reference levels. This proportion—60% of patients—is coincident according to the ESPEN and EASO Consensus [9,10].

Lean body mass was also evaluated following the recommendations of the ESPEN and EASO Consensus Statement for Sarcopenic Obesity and could diagnose 83.3% of our patients as mild or class-1 sarcopenia–with skeletal muscle mass (SMM)/weight levels below 1 SD–and 33.3% of the patients as severe or class-2 sarcopenia, with an SMM/weight level less than 2 SD.

According to both criteria [9]–excess fat mass and decreased lean mass–the prevalence of sarcopenic obesity was 56.7% (23.4% class-1 and 33.3% class-2) in our population.

### 3.4. Ultrasound Evaluation of RF Muscle

Postcritical COVID-19 patients had RF-CSA, 4.76 cm^2^ (3.56–5.43) for males and 3.65 cm^2^ (2.80–3.89) for females and were standardized by height 2.65 cm^2^/m (2.11–3.16) for males and 2.24 cm^2^/m (1.67–2.32) for females. The median muscle thickness (or Y-axis) was 1.38 cm (1.15–1.61), 1.40 cm (1.15–1.63) for males and 1.22 cm (0.87–1.60) for females and showed a strong positive correlation with HGS when evaluated by dynamometry (r = 0.559, *p* = 0.001). Qualitative parameters concerning the quality of the muscle were evaluated by the B-mode grayscale pixel values. The median of the mean echo intensity was 73.28 (50.69–81.8), while the median of the minimum echo intensity was 12 (0.25–27), and the median of the maximum echo intensity was 184 (176.25–203.75) of our population, without differences according to sex. Other parameters were measured from the muscle and adipose tissue ultrasound and are shown in Table 3.

### 3.5. Functional Status Assessment

The global mean HGS was 24.5 kg (19.28–28.25): 28 kg (20–40) for males and 20 kg (15–20) for females; it was found that 46.7% of the subjects had an HGS by dynamometry below the 10th percentile [15].

The mean global TUG was 8.6 s (6.28–10.2): 7.81 s (6.09–10.2) for males and 9.5 s (8.3–9.9) for females; it was found that 30% of the subjects achieved a time greater than 10 s in the TUG.

### 3.6. Degree of Agreement between Body Composition Techniques (Bioelectrical Measurements and Muscle Ultrasound): Correlation of the New Ultrasound Values with the Validated Measurement of BIVA Parameters of COVID-19 Post-Critical Patients

We found a high degree of agreement and complementarity between the body composition estimates found by BIVA as a reference value technique and muscle ultrasound. The PhA, as a global parameter of cell health, was positively correlated with muscle thickness (r = 0.389, *p* = 0.34). Other correlations between the muscle mass measured by BIVA and ultrasound are shown in Figure 4.

### 3.7. Relationship between the Functional Aspects and Body Composition Techniques (Bioelectrical Measurements and Muscle Ultrasound)

Functional measurements by dynamometry and TUG show high correlations between the body composition measures evaluated by BIVA and ultrasound techniques. Figure 5.

### 3.8. Evaluation of Qualitative Characteristics of the Muscle, and Muscle Quality

A greater mean echogenicity reflects both cases of scleroses related to older patients [r = 0.396, *p* = 0.037], and the fatty infiltration of muscle steatosis, which was positively associated with greater use of dexamethasone [r = 0.398, *p* = 0.036] and negatively associated with bioelectrical parameters of muscle mass such as SMI [r = −0.398, *p* = 0.036], and ultrasound such as RF-CSA [r = −0.433, *p* = 0.021]. 

However, the minimum and maximum echogenicity provided information on the muscular quality of the patient and related a greater minimum echo intensity of the RF muscle with parameters that reported a greater FM; this included both bioelectrical parameters, such as the FM percentage [r = 0.558, *p* = 0.002], and ultrasound subcutaneous adipose tissue of the leg [r = 0.405, *p* = 0.032], as well as the total [r = 0.401, *p* = 0.034] and superficial [r = 0.441, *p* = 0.019] abdominal adipose tissue, and lower cell mass, such as a lower FFM percentage [r = −0.558, *p* = 0.002], and the muscle circumference of the RF [r = −0.512, *p* = 0.005]. A maximum echo intensity of the RF muscle was related to older age [r = 0.395, *p* = 0.038]; a higher muscle was measured by the FFM percentage [r = 0.433, *p* = 0.021], and lower fat mass was measured by BI, such as the FM percentage [r = −0.433, *p* = 0.021], and by ultrasound, such as the adipose tissue of the leg [r = −0.414, *p* = 0.028].

Additionally, we performed a multivariate regression analysis to determine the model that best explained the qualitative characteristics of the muscle. We found a model that explained 65.3% of the estimate of the echo intensity of the RF muscle (*p* = 0.004) and a model that explained 61.3% of the estimate of the maximum echo intensity of the RF (*p* = 0.024).

**RF echo intensity** = 26.901 + 0.329 × Age (y)–0.430 × Stress Hyperglycaemia during admission (No: 0/Yes: 1) + 0.282 × Dexamethasone treatment (No: 0/Yes: 1).

**RF maximum echo intensity** = 177.727 + 0.183 × Age (y)–0.376 × FM (Kg) + 0.361 × Dexamethasone treatment (No: 0/Yes: 1)–0.185 × TUG test (s).

### 3.9. Establishment of Muscle Mass Estimation Algorithms and Cut-Off Value for Sarcopenia Diagnosis in Post-Critical COVID-19 Outpatients

Finally, based on the correlation analysis results, we performed a multivariate regression analysis to determine a predictive model that best explained the acquisition of SMI as the criteria for sarcopenia by using RF-CSA and the muscle thickness ultrasound parameters and HGS by dynamometry Table 4.

For the estimation of sarcopenia, predictive models with the highest diagnostic accuracy were established. The RF-CSA and muscle thickness ultrasound prediction models are algorithms 1 and 2, and the HGS prediction model is algorithm 3. The R^2^ values were 0.792, 0.774, and 0.856 for algorithms 1, 2, and 3, respectively. The standard error of the estimate (SEE) values were 1.1, 1.14, and 0.92 kg/m^2^ for algorithms 1, 2, and 3, respectively. Table 4.

To examine the capacity of the ultrasound measurement to predict sarcopenia in our population using ultrasound, we performed ROC analysis, and the AUC was 0.792 (95% CI: 0.613, 0.971, *p* = 0.007). Using the Youden index, which gives equal weight to false positive and false negative values, the optimum cut-off point for the maximum efficiency was 5.3 cm^2^/kg × 100 in RF-CSA/weight assessed by ultrasound (sensitivity = 88.2% and specificity = 69.2%. Figure 6.

## 4. Discussion

The authors should discuss the results and how they can be interpreted from the perspective of previous studies and of the working hypotheses. The findings and their implications should be discussed in the broadest context possible. Future research directions may also be highlighted.

The objective of our study was to evaluate sarcopenia and its risk factors in post-critical COVID-19 outpatients and to determine the validity and usability of the parameters obtained by nutritional ultrasound in the evaluation of the body composition of patients with severe COVID-19 who required admission to the ICU.

To the best of our knowledge, this is the first study that combines RF muscle ultrasound, BIVA, and functional measurements in the same cohort of COVID-19 outpatients to estimate parallel muscle mass and to describe the characteristics of the sarcopenic subset with this pathology.

The association of malnutrition and obesity with impaired measures of quantity and quality of muscle mass and fat mass distribution is very frequent in severe COVID-19 [16]. Our patient sample is representative of the COVID-19 population with severe disease and critical care, as it presents a high percentage of obese patients (greater than 63.3% according to a BMI > 30; 76.7% with an FM percentage 1SD above and 60% and with an FM percentage greater 2SD), which is related in the literature to a longer stay in the ICU since obesity is a factor risk of hospitalization and disease severity [3,17,18]. The muscle involvement component is more difficult to assess due to the BMI of these patients, so it may be hidden malnutrition. There is little evidence to provide information about the impact of body composition on the severity and outcomes of COVID-19 [19].

Therefore, this observational study showed that more than half of our population who were severe COVID-19 survivors had sarcopenic obesity following the new criteria for sarcopenia in obesity [9] established by the BIA, and nearly half of our population had a functional impairment of HGS by dynamometry.

Moreover, both the Global Leadership Initiative of Malnutrition guideline (GLIM criteria) [20] and the ESPEN and EASO Consensus Statement for Sarcopenic Obesity [9] recommended the evaluation of FFM and other muscle parameters (SMI) by BIA as a diagnostic criterion for malnutrition and sarcopenic, respectively. Based on the guideline recommendations, our study tried to evaluate sarcopenia in post-critical COVID-19 outpatients using new techniques such as ultrasound.

Muscle ultrasound is an emerging technique in nutritional assessment, which in recent years, has experienced significant development. A growing number of studies on this technique have been published. Additionally, muscle ultrasound has been used in ICU patients and has been described as a useful tool in the assessment of muscle atrophy in this situation [21,22,23].

In 2016, Mueller et al. [7] first proposed sex-adjusted RF-CSA defined by ultrasound as a tool to identify sarcopenic patients (43.1% of their cohort) and to predict adverse complications discharge/hospital mortality (OR 7.49, 95% CI:1.47–38.24; *p* = 0.015) and longer hospital stays in a cohort of surgical ICU patients. This result highlighted the potential of ultrasound-defined RF-CSA as a marker of complications and mortality, leading to the identification of sarcopenic patients [24].

Consequently, in a later work, Damanti et al. [25] analyzed a COVID-19 survivor cohort, proving that the ultrasound-defined muscle thickness of medial gastrocnemius could be an innovative tool to assess muscle mass in this population. Andrade-Junior et al. [2] demonstrated, in a critical COVID-19 patient series, that a loss of muscle mass occurred in the first 10 days of stay in the ICU. They showed a reduction of 30.1% in RF-CSA and 18.6% in muscle thickness. Umbrello et al. [26] also showed in critical COVID-19 patients, a loss of muscle mass occurred in the first 7 days of their ICU stay. They revealed a change in RF-CSA of less than 17.9% for survivors and less than 36.3% for non-survivor COVID-19 patients, with an RF-CSA median of 2.98 (2.17–3.97) and 2.49 (2.04–3.34), respectively. A reduced ultrasound-defined muscle mass has been associated with other poor clinical outcomes [27]. These findings are relevant in the evaluation of outpatients with severe COVID-19 after discharge from the ICU since there has been a very significant loss of muscle mass related to a hospital stay and the severity of the condition. In our study, ICU stay is a key aspect, together with the use of mechanical ventilation and age, for constituting risk factors that determine the muscle mass of these patients. There are no published reports on the evolution and recovery of muscle mass at hospital discharge in patients with severe COVID-19. Our study reflects a high prevalence of muscle involvement in these outpatients evaluated by BIVA. RF-CSA in our cohort compared to the Umbrello et al. series [26], showing the ambulatory recovery capacity of these patients.

Andrade-Junior et al. [2] showed that the use of corticosteroid therapy was related to parameters, such as hospital stay, HGS, or respiratory parameters. However, they did not find a correlation with echogenicity ultrasound during the early course of mechanical ventilation, while in our cohort, we found a correlation between the use of corticosteroid therapy and higher echo intensity. The echo intensity of muscle is a marker of muscle quality. Therefore, we established that clinical parameters such as corticosteroid therapy, stress hyperglycaemia, age, functional status, or level of fat mass, are the key to predicting the quality or echo intensity of muscle in these patients. This echo intensity increases in the context of inflammation, infection, edema, or corticoid treatment during ICU admission [28]. The echo intensity of the muscle could be an important clinical measure to be considered in the assessment of those patients that have survived critical illness [29]. In the present study, the median echo intensity showed similar values to a previous study by Umbrello et al. at 84.9 (75.4–94.9) in ICU survivors with COVID-19 on admission day [26]. Likewise, a recent longitudinal investigation showed that severe and critical COVID-19 patients showed an average 16.8% increase in echo intensity from days 1 to 10 [2]. Changes in muscle echogenicity have been associated with negative outcomes [29].

Additionally, this study has established that sarcopenia diagnosis can be estimated by algorithms including ultrasound measurements and HGS when adjusted by sex, age, and BMI. We propose that ultrasound can be an efficient method for the diagnosis of sarcopenia in post-critical COVID-19 outpatients. This fact has also been explored in elderly people with sarcopenia, with ultrasound proving to be a good tool for this purpose [30]. Abel et al. [31] found that combining the muscle thickness measured by ultrasound at eight sites throughout the body could predict the total body muscle mass with an error of 1.13 Kg in older people [31], and another paper on these groups showed that the error of prediction, using a single site on the forearm, was 1.95 kg [32]. Fukumoto et al. [33] found that the muscle thickness measured in the calf could predict low SMI better than in the quadriceps. However, Tang et al. [30] showed that the muscle thickness measured in the rectus femoris and vastus intermedius was the best site for estimating SMI in older people, with an error of 0.701 and 0.519 (for females and males, respectively). Our study covers this aspect for the first time in convalescent COVID-19 patients, and our results show that the prediction of total muscle mass in this population with an error of 1.10 or 1.14 used RF-CSA and muscle thickness, respectively. Moreover, some studies have also found that there may be differences in the rate of muscle atrophy at different sites, diverse pathologies, and between patients of different races [34,35,36].

Our results suggest the usefulness of ultrasound muscle measurements in the diagnosis of sarcopenic obesity in clinical practice. In our study, cut-off points of an RF-CSA/weight were less than 5.3 cm^2^/kg × 100 for the diagnosis of sarcopenia in obese outpatients with COVID-19. Deng et al. [37] studied skeletal muscle in patients with chronic obstructive pulmonary disease (COPD) and showed the predictive power of RF-CSA (AUC: 0.816, *p* < 0.001) in the diagnosis of sarcopenia.

## 5. Limitations and Strengths

This work presents some limitations. The small sample size (*n* = 30) and the bias were introduced due to the need for both the diagnostic ultrasound and BIA of patients enrolled in the study. In the distribution by gender, there is a clear predominance of males; however, there are no significant differences in the ultrasound muscle parameters when adjusted for height or weight.

Finally, independently of the precision of the ultrasound technique, the main advantage is the availability and efficiency. Ultrasound equipment is accessible in almost all medical centers. In addition, the ultrasound technique is low-cost, has bedside feasibility (during outpatient consultation or hospital admission), is completed quickly (about 5 min), does not involve biological risks, and can be performed at different patient follow-up points. In addition to these aspects, the future development of ultrasound to evaluate muscle and adipose tissue can provide qualitative data of great importance on muscle quality and related clinical events in patients (functional evaluation and metabolic impact of treatments).

## 6. Conclusions

Nutritional ultrasound, when applied to the assessment of body composition in post-critical patients with severe COVID-19, provides the possibility of assessing sarcopenia and obesity in these patients. The future usefulness of this ultrasonographic technique is based on the predictive value of sarcopenia diagnosis when estimating muscle mass quantitatively (RF-CSA and muscle thickness) and its qualitative characteristics (echo intensity), which reflect the different clinical circumstances of the patient and prognostic factors. The establishment of cut-off points for sarcopenia in patients with obesity is of great value when planning therapeutic strategies for the patient’s recovery. Further prospective and multi-center studies will be needed to define shared cut-offs to apply to the definition of sarcopenic obesity and to properly assess the possible differences in terms of sensitivity between such techniques.

## Figures and Tables

**Figure 1 nutrients-14-04988-f001:**
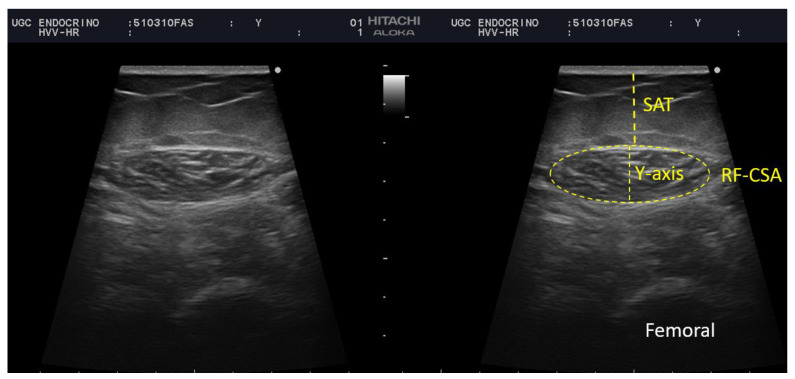
Measurement of rectus femoris (RF) and subcutaneous adipose tissue at leg level by ultrasound of a participant of our sample. BRF-CSA—cross-sectional area (CSA), muscle thickness (or Y-axis), SAT—subcutaneous adipose tissue.

**Figure 2 nutrients-14-04988-f002:**
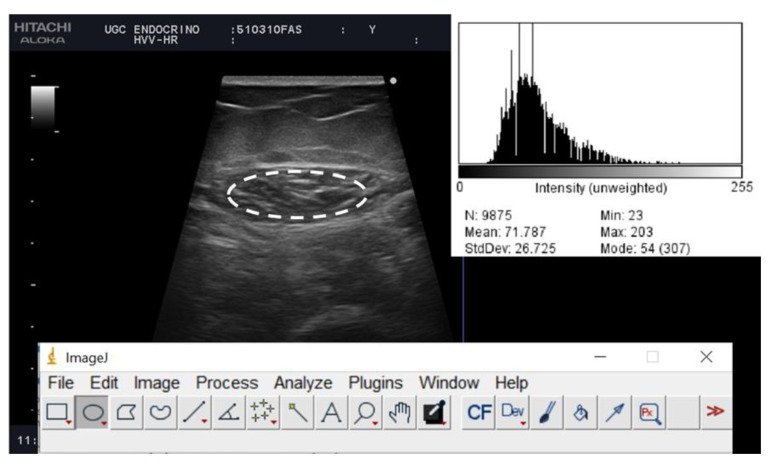
Muscle ultrasound images of post-critical COVID-19 outpatients from a representative patient. The histogram of the echo intensity of the rectus femoris (RF) is shown in the top right corner of the picture. A region of interest (rOI) is selected within the borders of the RF muscle (white line), and the echo intensity is determined using a histogram. Inset is the mean echo intensity of the rOI.

**Figure 3 nutrients-14-04988-f003:**
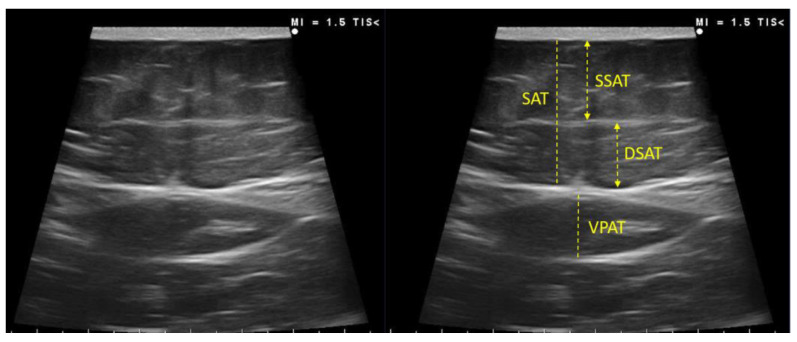
Measurement of subcutaneous and visceral preperitoneal adipose tissue at the abdominal level by ultrasound of a participant of our sample. SSAT—superficial subcutaneous adipose tissue; DSAT—deep subcutaneous adipose tissue; VPAT—visceral preperitoneal adipose tissue.

**Figure 4 nutrients-14-04988-f004:**
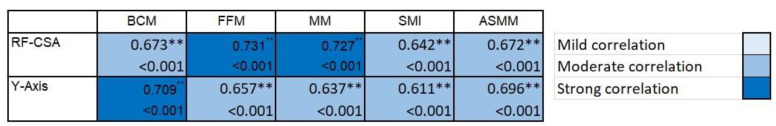
Correlation of the rectus femoris (RF) measurement by ultrasound and bioelectrical measurement in post-critical COVID-19 outpatients. In the heat map, the lightest blue represents a correlation coefficient of less than 0.39, and the darkest blue color represents a correlation coefficient of 0.7; the larger the correlation coefficient, the darker the corresponding color. ** *p* < 0.01.

**Figure 5 nutrients-14-04988-f005:**
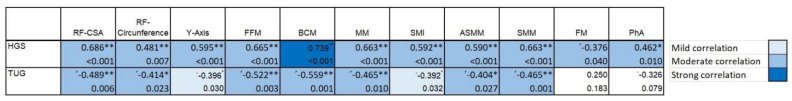
Correlation of the body composition techniques (bioelectrical and ultrasound measurements) and functional aspects (HGS and TUG) in post-critical COVID-19 outpatients. In the heat map, the lightest blue represents a correlation coefficient of less than 0.39, and the darkest blue color represents a correlation coefficient of 0.7; the larger the correlation coefficient, the darker the corresponding color. * *p* < 0.05; ** *p* < 0.01.

**Figure 6 nutrients-14-04988-f006:**
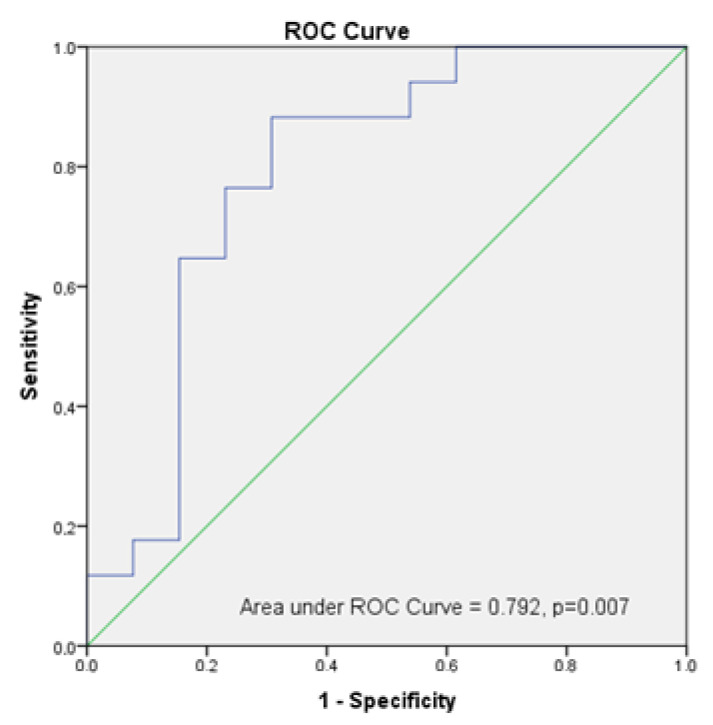
Analysis of ROC Curve of RF-CSA/weight to predict sarcopenia in post-critical COVID-19 outpatients.

**Table 1 nutrients-14-04988-t001:** Baseline demographic characteristics of post-critical COVID-19 outpatients.

	Participants (*n* = 30)
Age (years)	60 ± 9.4
Male *n* (%)	23 (76.7)
BMI (kg/m^2^)	31.6 ± 7.4
ICU stay (days)	10 ± 16.5
Hospital stay (days)	23 ± 19.9
Stress hyperglucemia (%)	73.3
Mechanical Ventilation (%)	53.3
Manoeuvres prone (%)	46.7
Corticosteroid Therapy (%)	83.3
Home oxygen therapy after hospital admission (%)	53
*Comorbidities*	
Diabetes Mellitus (%)	26.7
Arterial Hypertension	53.3
Dyslipidaemia (%)	43.3
Obesity by BMI ≥ 30 kg/m^2^(%)	63.3

BMI—body mass index; ICU—intensive care unit.

**Table 2 nutrients-14-04988-t002:** Multiple linear regression analysis of clinical risk factors during admission of sarcopenia based on skeletal muscle index (SMI) in post-critical COVID-19 outpatients.

Independent Variables	Standardized β	95% CI	*p*
Length of ICU stay	0.636	0.026, 0.148	0.007 *
Mechanical ventilation	−0.554	−4.308, −0.594	0.012 *
Age	−0.323	−0.149, −0.006	0.035 *
Sex	−0.260	−3.048, 0.340	0.112

* ICU—intensive care unit, CI—confidence interval.

**Table 3 nutrients-14-04988-t003:** Ultrasound evaluation of rectus femoris (RF) muscle and adipose tissue in post-critical COVID-19 patients.

		Overall Median (Interquartile Range)	Male (n = 23) Median (Interquartile Range)	Female (n = 7) Median (Interquartile Range)	*p* ª
	**MUSCLE ASSESSMENT**				
	*Quantitative parameters*				
	Cross-sectional area (cm^2^)	4.35 (3.5–5.33)	4.76 (3.56–5.43)	3.65 (2.80–3.89)	0.025
	Cross-sectional area/height (cm^2^/m) Cross-sectional area/weigh (cm^2^/Kg)	2.54 (2.07–3.04) 0.050 (0.039–0.060)	2.65 (2.11–3.16) 0.051 (0.041–0.060)	2.24 (1.67–2.32) 0.037 (0.026–0.060)	0.069 0.065
	Muscle circumference	9.43 (8.53–10.15)	9.54 (8.73–10.30)	8.61 (7.55–9.83)	0.096
	Muscle circumference/height (cm^2^/m)	5.40 (4.93–5.93)	5.42 (5.02–5.90)	5.25 (4.58–6.03)	0.532
	Muscle thickness (Y-axis)	1.38 (1.15–1.61)	1.40 (1.15–1.63)	1.22 (0.87–1.60)	0.564
	X-axis	3.95 (3.35–4.24)	3.99 (3.66–4.24)	3.28 (2.69–4.48)	0.266
	*Qualitative parameters*				
	Mean Echo intensity	73.28 (50.69–81.8)	69.09 (47.60–78.09)	79.11 (56.98–93.15)	0.296
	Minimum echo intensity	12 (0.25–27)	12 (0–24.5)	12 (5–34)	0.321
	Maximum echo intensity	184 (176.25–203.75)	187 (177–207.5)	177 (170–196)	0.090
Leg	**ADIPOSE TISSUE ASSESSMENT**				
	Subcutaneous Adipose tissue	0.76 (0.50–1.49)	0.71 (0.47–0.83)	1.66 (1.21–1.80)	0.007
Abdomen	Total subcutaneous adipose tissue	1.80 (1.30–2.60)	1.65 (1.02–2.35)	2.98 (1.75–3.72)	0.008
	Superficial subcutaneous adipose tissue	0.84 (0.57–1.27)	0.75 (0.42–0.92)	1.52 (0.99–2.09)	0.002
	Visceral preperitoneal adipose tissue	0.70 (0.42–0.87)	0.70 (0.43–0.86)	0.69 (0.38–0.90)	0.917

ª for the comparison of male and female.

**Table 4 nutrients-14-04988-t004:** Skeletal muscle index (SMI) estimated algorithms as criteria for sarcopenia based on ultrasound assessment of rectus femoris (RF).

SKELETAL MUSCLE INDEX (SMI) ESTIMATED				
**Algorithm 1 (Kg/m^2^)**	**R**	**R^2^**	**SEE (Kg/m^2^)**	** *p* **
Estimated SMI = 1.015 + 0.246 × RF-CSA (cm^2^) + 0.714 × BMI (Kg/m^2^)–0.433 × Sex (male:0/female:1) + 0.45 × Age (y).	0.890	0.792	1.10	<0.001
**Algorithm 2 (Kg/m^2^)**	**0.880**	**0.774**	**1.14**	**<0.001**
Estimated SMI = 1.006 + 0.193 × Y-axis (cm) + 0.711 × BMI (Kg/m^2^)–0.499 × Sex (male:0/female:1) + 0.08 × Age (y).				
**Algorithm 3 (Kg/m^2^)**	**0.925**	**0.856**	**0.92**	**<0.001**
Estimated SMI = −1.584 + 0.369 × HGS (Kg) + 0.820 × BMI (Kg/m^2^)–0.403 × Sex (male:0/female:1) + 0.155 × Age (y).				

SEE—standard error of estimated; SMI—skeletal muscle index; RF-CSA—rectus femoris cross-sectional area; BMI—body mass index; HGS—handgrip strength.

## Data Availability

Data described in the manuscript, code book, and analytic code will be made available upon request to the corresponding author.

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
