# Peer review of "Predictors of Sarcopenia in Outpatients with Post-Critical SARS-CoV2 Disease. Nutritional Ultrasound of Rectus Femoris Muscle, a Potential Tool"

_nutrients, 2022, doi:10.3390/nu14234988_

Round 1

Reviewer 1 Report

I suggest minor changes

97- what kind of exercises for muscle recovery? And what was the protein supplementation recommended in the protocol?

 208- lack of BMI reference.

3.2 242- Not only does low muscle determine sarcopenia but also reduced grip strength and functionality251- Is it sarcopenic obesity?

Comment: As for the ultrasound evaluation protocol, it seems to me to be adequate, although time-consuming and technically demanding (it needs professionals in the area) what is not specified in the study

Author Response

Dear Editor and Reviewers,

We would like to thank you very much for your constructive comments and suggestions which have undoubtedly helped us to improve our manuscript.

We have taken these comments and suggestions into consideration and have revised the paper accordingly. We have made all possible efforts to respond to each of the reviewers’ comments and have edited the manuscript where we were able to fully address the reviewers’ suggestions.

We have provided the replies to the comments in the following section and have highlighted changes in the manuscript in red font.

We hope that our revised manuscript may now be found acceptable for publication in the journal. Nevertheless, we are of course willing to revise it further according to any other suggestions or concerns raised by the Editor or the Reviewers.

Yours faithfully,

Isabel Cornejo-Pareja,

Reviewer 2 Report

I have read a transversal study regarding a relatively new tool of assessing sarcopenia in patients that were critically affected by SARS-CoV2 disease. The authors wrote a nice manuscript, with good insights on the pathology.

Abstract

Row 19 Please replace "Objetives" with "Objectives:"

Row 27, 28, 29. There are some abbreviations like ESPEN, etc. Please define first before use.

When pointing percentages of patients, it is recommended to add their number too; e.g. 46.1% (n=xx)

Manuscript

I suggest the row 54 and 55 to be the same paragraph as they are both describing BIVA.

Add a period at the end of row 71.

Row 102, 131, 132, 163 - is this a subtitle? Is this a sub-sub-title? Please add numerical values to each section of your subsections.

Row 182 - can you specify which function was assessed?

Table 1 - please add standard deviation to all the values that have a range. Replace this in text too.

There are many misplaced citations in the text e.g. criteria[9]; ESPEN and EASO Consensus.[9,10] etc. Please place the citation with a space after your last word, and also before the final period of each sentence (and not after, like in the example). This is a general rule.

The ethical approval number and date should be added at the end of your manuscript (not necessarily in the methods section).

 The reference style and their novelty is ok, with no issues.

Author Response

(The authors gave the same response as above.)
